# Antioxidant Potential of Non-Extractable Fractions of Dried Persimmon (*Diospyros kaki* Thunb.) in Streptozotocin-Induced Diabetic Rats

**DOI:** 10.3390/antiox11081555

**Published:** 2022-08-11

**Authors:** Naoko Mochida, Yoko Matsumura, Masahiro Kitabatake, Toshihiro Ito, Shin-ichi Kayano, Hiroe Kikuzaki

**Affiliations:** 1Department of Food Science & Nutrition, School of Humanities & Science, Nara Women’s University, Nara 630-8506, Japan; 2Department of Health and Nutrition, Faculty of Health Science, Kio University, Kitakatsuragi-gun, Nara 635-0832, Japan; 3Department of Immunology, Nara Medical University, Kashihara City 634-8521, Japan; 4Department of Food Science & Nutrition, Nara Women’s University, Nara 630-8506, Japan

**Keywords:** dried persimmon, diabetes, oxidative stress, liver, muscle fibers

## Abstract

Oxidative stress causes the progression of diabetes and its complications; thus, maintaining the balance between reactive oxygen species produced by hyperglycemia and the antioxidant defense system is important. We herein examined the antioxidant potential of non-extractable fractions of dried persimmon (NEP) against oxidative stress in diabetic rats. Rats with streptozotocin-induced type 1 diabetes (50 mg/kg body weight) were administered NEP for 9 weeks. Antioxidant enzyme activities and concentration of antioxidants in liver tissues were analyzed with a microplate reader. Extensor digitorum longus (EDL) and soleus muscle fibers were stained with succinate dehydrogenase and muscle fiber sizes were measured. The administration of NEP increased the body weight of diabetes rats. Regarding antioxidant activities, the oxygen radical absorbance capacity and superoxide dismutase activity in liver tissues significantly increased. In addition, increases in glutathione peroxidase activity in liver tissues and reductions in the cross-sectional area of EDL muscle fibers were significantly suppressed. In these results, NEP improved the antioxidant defense system in the liver tissues of diabetic rats, in addition to attenuating of muscle fibers atrophy against oxidative damage induced by hyperglycemia.

## 1. Introduction

Fruits and vegetables are rich in vitamins, minerals, fiber, and natural antioxidants. The sustained intake of fruits and vegetables has been shown to reduce the risk of diseases, such as cancer, hypertension, and cardiovascular diseases [1,2,3]. Persimmon (*Diospyros kaki* Thunb.) belongs to the Ebenaceae family and is rich in vitamins, minerals, fiber, and bioactive compounds. Persimmon is generally classified into astringent or non-astringent varieties. The fruit of astringent persimmon is rich in soluble condensed tannin [4] and, thus, has a very bitter taste. However, soluble tannin is converted into insoluble tannin during the drying process, and, as a result, dried persimmon loses its bitterness and develops a sweet taste. Condensed tannin has been shown to exhibit high antioxidant activity [5,6]. Previous studies demonstrated that tannin in persimmon reduced blood pressure [7], exhibited antibacterial activities [6,8], and exerted hypolipidemic effects [9,10]. Furthermore, persimmon was found to exert different beneficial effects against diseases, such as hypertension, diabetes mellitus, and atherosclerosis [11].

Diabetes mellitus is a major health issue worldwide. Over 10.5% of the world’s population suffers from diabetes, which is projected to increase to 12.2% by 2045 [12]. It causes hyperglycemia due to deficiencies in insulin secretion and/or insulin action. The oxidation of glucose, the excess production of superoxide anions in mitochondria, and the non-enzymatic glycation of proteins occur under hyperglycemic conditions [13,14,15], which promote oxidative stress [16]. Oxidative stress occurs when pro-oxidant levels exceed those of antioxidants in vivo [17]. Cells are constantly exposed to oxidative stress. Typical reactive oxygen species (ROS) include superoxide, hydrogen peroxide (H_2_O_2_), hydroxyl radical, singlet oxygen, NO, etc. They react to DNA, proteins, lipids, etc., and cause dysfunction. ROS are also produced by environmental factors, such as smoking, UV light, radiation, and air pollution [18]. Increased oxidative stress contributes to the progression of diabetes and its complications. The sustained production of ROS by hyperglycemia plays an important role in the development of damage associated with diabetes [19,20,21]. On the other hand, organisms have an antioxidant defense system against ROS and demonstrate antioxidant activities by eliminating ROS or converting them to other substances. Antioxidant enzymes include superoxide dismutase (SOD), catalase, and glutathione peroxidase (GPx). Glutathione, thioredoxin, vitamin C, vitamin E, cysteine, uric acid, bilirubin, etc., are known as antioxidants, and albumin, ferritin, etc., are known as antioxidant proteins [18]. The maintenance of a balance between ROS production and the antioxidant defense system is the main mechanism responsible for preventing oxidative stress-induced damage. Therefore, antioxidants may be helpful in the treatment of diabetes.

The antioxidant activities of fruits are generally evaluated using their soluble extracts, but few assessments have been conducted on non-extractable fractions. However, it is possible that the non-extractable fraction of dried persimmon (NEP) is rich in condensed tannin. We previously demonstrated that NEPs exhibited high antioxidant activity in vitro and in vivo [22]. Therefore, the aim of the present study was to investigate the antioxidant potential in NEP against oxidative stress in streptozotocin-induced diabetic rats.

## 2. Materials and Methods

### 2.1. Chemicals and Apparatus

Trolox (6-hydroxy-2,5,7,8-tetrametylcfroman-2-carboxylic acid) and fluorescein sodium salt were purchased from Sigma-Aldrich Japan Co., Ltd. (Tokyo, Japan). 2,2′-Azobis (2-amidinopropane) dihydrochloride (AAPH) was obtained from FUJIFILM Wako Pure Chemical Corp. (Osaka, Japan). All reagents were of analytical grade.

HPLC analysis was performed using a Shimadzu LC-10AD VP multisolvent delivery system equipped with a SIL-10AD VP auto injector and SPD-M10A VP photodiode array detector (Shimadzu Corp., Kyoto, Japan). A Diaion HP-20 (Mitsubishi Chemical Co., Ltd., Tokyo, Japan) was used for the column chromatography. Oxygen radical absorbance capacity (ORAC), SOD, and GPx activities, and concentrations of total glutathione and oxidized glutathione (GSSG) were measured using the PerkinElmer 2030 Multilabel Microplate Reader ARVO^TM^ X4 (PerkinElmer Japan Co., Ltd., Kanagawa, Japan). Absorbance measurements of the concentration of H_2_O_2_ remaining after the effects of catalase were performed using the Shimadzu UV mini-1240 spectrophotometer (Shimadzu Corp., Kyoto, Japan).

### 2.2. Plant Material

The “Hohlenbo” variety of astringent persimmon (*Diospyros kaki* Thunb.) was harvested in Yoshino-gun, Nara, Japan. Fruit was peeled and sulfureted to prevent discoloration and oxidation and then processed into dried persimmon after air-drying for one month.

### 2.3. Preparation of NEP

After the removal of seeds from dried persimmon, the pulp (2885.4 g) was cut into small pieces and homogenized with 14.8 L of 90% aqueous ethanol (EtOH). The extract was filtered from the homogenate and 90% aqueous EtOH was added to the residue. These extracting and filtering procedures were repeated three times. The extracted residue was dried at room temperature to obtain NEP (769.3 g).

### 2.4. Preparation of Persimmon-Derived Tannin

The procedure for persimmon-derived tannin extraction has been described previously [23,24,25]. Immature persimmon fruits (*Diospyros kaki* Thunb., cv. “Hiratane-nashi” and “Tone-wase”) harvested in Nara, Japan, were used for tannin extraction. “Hiratane-nashi” was designated as a protected species (No. 24, Niigata, Japan) in 1962. “Tone-wase”, a mutated variety of ”Hiratane-nashi”, was registered (“The Plant Variety Protection No. 28”, Ministry of Agriculture, Forestry and Fisheries, Japan) in 1980. The species of persimmon were identified by Dr. Sadahiro Hamasaki (Nara Prefecture Agricultural Research and Development Center, Nara, Japan), and the extraction of tannin was conducted using a modified method from his study [26]. Immature persimmon fruits were treated with 0.2% ethanol (*w*/*v*) for 5 days for the insolubilization of tannin. The ethanol-treated persimmons were crushed into small pieces and soaked in water for 2 days at room temperature. The supernatant contained soluble components, such as sugars, and the residue contained insoluble tannin. After discarding the supernatant, water was added to the residue. The suspension was heated at 120 °C for 30 min to exchange insoluble tannin for soluble tannin. The suspension was filtered, evaporated in vacuo, and dried at 160 °C to obtain a powder of persimmon-derived tannin. The batch of this powder contained 75.5% condensed tannin, measured as epigallocatechin gallate equivalent based on the Folin–Ciocalteau method. Persimmon-derived tannin was stored at −20 °C until used.

### 2.5. Fractionation of Hydrolyzed NEP and Persimmon-Derived Tannin

Portions (1 g) of NEP or persimmon-derived tannin were combined with 50 mL of a 1.2 M HCl-50% methanol (MeOH) solution and heated at 90 °C for 3 h. The treated suspensions were centrifuged at 3000 rpm at room temperature for 15 min to obtain the supernatant. The same solution was added to the residue, and they were heated again at 90 °C for 30 min. The suspensions were centrifuged under the same conditions to obtain another supernatant. The combined supernatant was subjected to Diaion HP-20 column chromatography using H_2_O as an eluting solution, followed by elution with 100% MeOH. The H_2_O eluate was neutralized with NaOH, and ethyl acetate (EtOAc) was added to the partition between EtOAc and H_2_O. The EtOAc layer was evaporated in vacuo to obtain the fractions of hydrolyzed NEP or persimmon-derived tannin for HPLC analysis.

### 2.6. HPLC Analysis of Hydrolyzed NEP and Persimmon-Derived Tannin

Each fraction of NEP or persimmon-derived tannin was dissolved in 2 mL of 100% MeOH and filtered through 0.45 µm disposable membrane filters (Toyo Roshi Kaisha, Ltd., Tokyo, Japan). The conditions of HPLC were as follows: column, RPAQUEOUS-AR-5, 4.6 × 250 mm, (Nomura Chemical Co., Ltd., Aichi, Japan); column temp., 40 °C; mobile phase, A = 0.1% formic acid, B = 0.1% formic acid in 30% acetonitrile; flow rate, 1.0 mL/min; gradient, 0.00 min, %B = 0; 5.00 min, %B = 20.0; 10.00 min, %B = 21.0; 20.00 min, %B = 23.0; 25.00 min, %B = 25.0; 30.00 min, %B = 5.0; 35.00 min, %B = 0; 40.00 min, %B = 0; injection volume, 5µL; detection, photodiode array (190–600 nm).

### 2.7. Animal Studies

#### 2.7.1. Animal and Feeding Procedures

Type 1 diabetes was induced in eight-week-old male Wistar rats with streptozotocin (50 mg/kg body weight) and normal rats were obtained from Japan SLC (Hamamatsu, Shizuoka, Japan). Normal rats were fed an AIN-93G-modified basal diet (CLEA Japan Inc., Tokyo, Japan) (NOC group). Diabetic rats were randomly divided into two groups of six animals each, as follows: (i) diabetic rats fed an AIN-93G-modified basal diet (DMC group); (ii) diabetic rats fed a basal diet supplemented with 5% NEP instead of cellulose (DMP group). Each animal was individually housed. Study diet compositions are shown in Table 1.

Animals were fed *ad libitum*. Food intake and the body weights of all groups were monitored daily for 9 weeks. Blood was collected from the tail vein weekly, and plasma was isolated and stored at −80 °C until use. Rats were killed after 9 weeks, and the liver and skeletal muscle tissues (extensor digitorum longus (EDL) and soleus muscles) were isolated. Isolated liver tissues were perfused with ice-cold physiological saline, immediately frozen in a dry-ice acetone bath, and stored at −80 °C until use. Isolated muscles were frozen in liquid N_2_ and stored at −80 °C until use.

#### 2.7.2. ORAC Assay of Plasma

Plasma samples were removed from storage at −80 °C and thawed under running water. Aliquots (50 µL) were inserted into tubes and 100 µL of EtOH and 50 µL of H2O were added. After shaking for 30 s using a vortex, 200 µL of 0.75 M metaphoric acid was added. The mixtures were centrifuged at 210× *g* at 10 °C for 5 min. The supernatants were diluted 1:6.25 (*v*/*v*) in 75 mM phosphate buffer (pH 7.4) to obtain the plasma solution. Plasma solutions were further diluted two- to eightfold in 75 mM phosphate buffer (pH 7.4).

The ORAC of each sample was assessed according to a previously reported method [27,28,29,30] with slight modifications. ORAC assays were performed at an excitation wavelength of 485 nm and emission wavelength of 535 nm.

ORAC was measured at 37 °C every 2 min for 90 min. The area under the fluorescence curve was calculated, and the ORAC of each sample was expressed as units for 1 µmol equivalent of Trolox.

#### 2.7.3. ORAC Assay of Liver Tissues

Liver tissues were weighed to approximately 300 mg and added to 10 volumes (*w*/*v*) of PBS (pH 7.5) (Dojindo Morecular Technologies, Inc., Kumamoto, Japan) with 0.5% Triton X-100 (Fujifilm Wako Pure Chemical Corp., Osaka, Japan). Suspensions were then homogenized, and the resulting homogenates were centrifuged at 22,360× *g* at 4 °C for 10 min. Supernatants were stored at −80 °C until use. Prior to the ORAC assay, supernatants were thawed under running water and further diluted 32-fold in 75 mM phosphate buffer (pH 7.4). The ORAC of diluted supernatant solutions was measured as described above (Section 2.7.2).

#### 2.7.4. SOD Activity Assay

Liver tissues were weighed to approximately 100 mg and added to 10 volumes (*w*/*v*) of 20 mM HEPES buffer (pH 7.2) (Dojindo Morecular Technologies, Inc., Kumamoto, Japan) with 1 mM EGTA (Kanto Chemical., Co., Ltd., Inc., Osaka, Japan), 210 mM mannitol (Fujifilm Wako Pure Chemical Corp., Osaka, Japan), and 70 mM sucrose (Fujifilm Wako Pure Chemical Corp., Osaka, Japan). Suspensions were then homogenized, and the resulting homogenates were centrifuged at 1500× *g* at 4 °C for 5 min. Supernatants were further centrifuged at 14,000× *g* at room temperature for 15 min to concentrate the samples using an Amicon concentrator (Merck Millipore Ltd., Tokyo, Japan) with a molecular weight cut-off of 10,000. The samples obtained were stored at −80 °C until used. Prior to the SOD activity assay, filtrates were thawed under running water.

SOD activity was assessed using a previously reported method [31]. Activity was measured using a commercial ELISA kit (Superoxide Dismutase Assay Kit, Cayman chemical, Ann Arbor, MI, USA, inter-assay coefficient of variation 3.7%, dynamic range 0.005–0.05 units/mL SOD). This assay was performed according to the manufacturer’s instructions. One unit of SOD was defined as the amount of enzyme needed to exhibit 50% dismutation of the superoxide radical.

#### 2.7.5. Catalase Activity Assay

Liver tissues were weighed to approximately 100 mg and added to 350 µL of PBS with 0.5% Triton X-100 and 1% protease inhibitor (Fujifilm Wako Pure Chemical Corp., Osaka, Japan). Suspensions were then pulverized with the Micro Smash Ms-100R bead beater (TOMY Japan) at 4500 rpm at 4 °C for 30 s. The resulting solutions were centrifuged at 11,000× *g* at 4 °C for 1 min. Supernatants were transferred into other tubes and centrifuged at 11,000× *g* at 4 °C for 15 min. The resulting supernatants were stored at −80 °C until use. Before the catalase assay, supernatants were thawed under running water and further diluted 150-fold in 50 mM potassium phosphate buffer (pH 7.0).

Catalase activity was assessed using a previously reported method [32,33] with a commercial kit (Catalase Assay Kit, Sigma Aldrich, Missouri, America. This assay was conducted according to the manufacturer’s instructions. Catalase activity was calculated from the concentration of H_2_O_2_ remaining after the effects of catalase. One unit of catalase was expressed as units required for 1 µmol of H_2_O_2_ to decompose to oxygen and water per minute with a substrate concentration of 10 mM H_2_O_2_.

#### 2.7.6. GPx Activity Assay

Liver tissues were weighed to approximately 400 mg and added to 5 volumes (*w*/*v*) of 50 mM Tris-HCl buffer (pH 7.5) (Nippon Gene Co., Ltd., Tokyo, Japan) with 5 mM EDTA (Fujifilm Wako Pure Chemical Corp., Osaka, Japan) and 1 mM 2-mercaptoethanol (Fujifilm Wako Pure Chemical Corp., Osaka, Japan). Suspensions were then homogenized, and the resulting homogenates were centrifuged at 8000× *g* at 4 °C for 10 min. Supernatants were stored at −80 °C until used. Prior to the GPx activity assay, supernatants were thawed under running water. Supernatants were further diluted 10-fold in phosphate buffer with EDTA and 4 mM NaN3 (pH 7.0).

The GPx activity assay is an adaptation of the method developed by Paglia and Valentine [34]. Its activity was measured using a commercial kit (NWLSS^TM^ Glutathione Peroxidase Activity Assay Kit, Northwest Life Science Specialities LLC, Vancouver, WA, USA, sensitivity 5.0 mU/mL, dynamic range 5–65 mU/mL). This assay was performed according to the manufacturer’s instructions. One unit of GPx-1 was expressed as the amount of enzyme needed to catalyze the oxidation of 1 µmol reduced glutathione (GSH) to GSSG per minute.

#### 2.7.7. Measurement of Total Glutathione and GSSG Concentrations

Liver tissues were weighed to approximately 100 mg and added to 10 volumes (*w*/*v*) of 5% 5-sulfosalicylic acid solution (Fujifilm Wako Pure Chemical Corp., Osaka, Japan). Suspensions were then homogenized, and the resulting homogenates were centrifuged at 8000× *g* at 4 °C for 10 min to remove proteins. Supernatants were stored at −80 °C until used. Prior to the total glutathione and GSSG assay, supernatants were thawed under running water. Supernatants were then further diluted 50-fold in ultrapure water for total glutathione and 5-fold in ultrapure water for GSSG.

Total glutathione and GSSG concentrations were measured using a previously reported method [35,36] with a commercial kit (GSSG/GSH Quantification Kit, Dojindo, Kumamoto, Japan). This assay was conducted according to the manufacturer’s instructions. GSH was calculated from the concentrations of total glutathione and GSSG, and the ratio of GSH and GSSG was obtained as an index of oxidative stress. Total glutathione, GSH, and GSSG concentrations were normalized to the wet weight of the tissue sample.

#### 2.7.8. Measurement of Protein Concentrations in Plasma and Liver Tissues

Total protein concentrations were measured using the bicinchoninic acid (BCA) method with bovine serum albumin as a standard (TaKaRa BCA Protein Assay Kit, Shiga, Japan). The ORAC and SOD, catalase, and GPx activities of liver tissues were normalized to the protein concentrations of the homogenates.

#### 2.7.9. Measurement of Cross-Sectional Area (CSA)

To measure the CSA of EDL and soleus muscle fibers, 10 µm thick transverse sections of muscle were stained with succinate dehydrogenase (SDH) and examined under a light microscope (model BZ-9000, Keyence, Osaka, Japan). SDH staining is a method that uses the enzyme activity involved in energy production in mitochondria and classifies muscle fibers into slow-twitch oxidative (SO) plus fast-twitch oxidative glycolytic (FOG) and fast-twitch glycolytic (FG). Muscle fiber sizes were measured using ImageJ (version 1.46; National Institutes of Health, Bethesda, MD, USA).

### 2.8. Statistical Analysis

Each sample in all assays was measured in duplicate, and data were expressed as means ± standard deviations. Statistical analyses were performed using the Mann–Whitney U test and differences were considered to be significant at *p* < 0.05.

## 3. Results

### 3.1. HPLC Analysis of Hydrolyzed NEP and Persimmon-Derived Tannin

It was speculated that NEPs are rich in condensed tannin. Therefore, HPLC analysis was performed to confirm the existence of tannin in NEPs. HPLC chromatograms of hydrolyzed NEP and persimmon-derived tannin are shown in Figure 1. Two major peaks were detected in the NEP (Figure 1A) and each peak was also detected in persimmon-derived tannin at the same retention time (Figure 1B). The absorption spectra of these peaks were the same in both hydrolyzed fractions; therefore, the NEP contained condensed tannin with the same composition as persimmon-derived tannin.

### 3.2. Food Intake and Body Weight Gain in Rats

Food intake and body weight gain in the three groups of rats were monitored. No significant differences were observed in food intake in the three groups of rats. On the other hand, body weight was significantly higher in the NOC group (*p* < 0.001) than in the two other groups of diabetic rats (DMC and DMP groups). The body weight after 3 weeks was significantly higher in the DMP group than in the DMC group (*p* < 0.05); however, the body weight after 9 weeks did not significantly differ between DMC and DMP groups (Figure 2).

### 3.3. Effects of NEP on Antioxidant Activity in Plasma and Liver Tissues

The effects of NEP on the ORAC of the plasma and liver tissues of experimental rats are shown in Figure 3. Plasma ORAC after 9 weeks was significantly lower (*p* < 0.05) in the two groups of diabetic rats (DMC and DMP groups) than in the NOC group (Figure 3A). On the other hand, the ORAC of liver tissues after 9 weeks was significantly higher (*p* < 0.05) in the DMP group than in the two other groups (NOC and DMC groups) (Figure 3B).

The effects of NEP on antioxidant enzyme activity in experimental rats are shown in Figure 4. No significant differences were observed in the SOD activity of liver tissues after 9 weeks between the DMC group and NOC group; however, it was significantly higher (*p* < 0.05) in the DMP group than in the NOC and DMC group (Figure 4A). The catalase activity of liver tissues after 9 weeks did not significantly differ in the three groups of rats; however, it was slightly lower in the DMC group than in the NOC group, and it was slightly higher in the DMP group than in the DMC group (Figure 4B). The GPx activity of liver tissues after 9 weeks was significantly higher (*p* < 0.05) in the DMC group than in the NOC group; however, the increase observed in the DMP group was slightly smaller than that in the DMC group (Figure 4C). These results demonstrated that the administration of NEP enhanced SOD activity and suppressed increases in GPx activity in diabetic rats.

The effects of NEP on the GSH/GSSG ratio and total glutathione, GSH, and GSSG concentrations in experimental rats are shown in Figure 5. No significant differences were observed in the GSH/GSSG ratio or total glutathione, GSH, and GSSG concentrations in liver tissues after 9 weeks between the DMC and NOC groups. However, the GSH/GSSG ratio, and total glutathione and GSH concentrations were significantly lower (*p* < 0.05) in the DMP group than in the NOC and DMC groups (Figure 5A–C). GSSG concentrations in liver tissues after 9 weeks were significantly (*p* < 0.05) higher in the DMP group than in the NOC and DMC groups (Figure 5D). These results demonstrated that the administration of NEP to diabetic rats suppressed the GPx-glutathione system in liver tissues.

### 3.4. Effects of NEP on the Characterization of EDL and Soleus Muscle Fibers

Photomicrographs of cross-sections through the EDL muscle of each group stained with SDH are shown in Figure 6. The effects of NEP on the CSA of EDL and soleus muscle fibers in each group are shown in Figure 7. EDL muscle fibers were smaller in the DMC group than in the NOC group (Figure 6A,B); however, the EDL muscle in the DMP group was improved to the same extent as in the NOC group, as shown in Figure 6A,C. The CSA of all muscle fiber types in the EDL and soleus muscles were significantly smaller in diabetic rats. The CSA of SO fibers and FOG fibers in the soleus muscle did not significantly differ between the DMC and DMP groups (Figure 7B). The CSA of all muscle fiber types in the EDL muscle were significantly higher in the DMP group than in the DMC group (Figure 7A).

## 4. Discussion

Oxidative stress occurs when pro-oxidant levels exceed those of antioxidants in vivo [17]. Oxidative stress has been implicated in the pathogenesis of various diseases, such as obesity, diabetes mellitus, and aging [13,14,15,19,37,38].

The ORAC assay is a type of antioxidant potential assay that is commonly used to assess the radical (AAPH) scavenging activity of biological samples [27,28,29,30]. In the present study, the ORAC of plasma was significantly lower in the DMC group than in the NOC group (Figure 3A), which may reflect excess oxidation in the DMC group. No significant differences were observed in the ORAC of plasma between the DMC and DMP groups (Figure 3A). Furthermore, the ORAC of liver tissues did not significantly differ between the NOC and DMC groups but was significantly higher in the DMP group than in the NOC and DMC groups (Figure 3B). Therefore, ORAC of liver tissues may depend on the antioxidant potential in NEP. The discrepancy between the results obtained for plasma and the liver may be attributed to tissue specificity. Previous studies reported differences between the ORAC of plasma and liver tissues in kolavion-treated diabetic rats [21,39].

Oxidative stress markers are broadly classified into three types: antioxidant enzymes, antioxidants, and products produced in vivo by ROS. Antioxidant enzymes, including SOD, catalase, and GPx, remove ROS. SOD catalyzes the degradation of the superoxide radical anion. Catalase disproportionates H_2_O_2_ into water and oxygen. GPx detoxifies H_2_O_2_ and lipid peroxides [40]. When the production of ROS exceeds the capacity of antioxidant enzymes in vivo, antioxidants then become involved. Glutathione is the most abundant intracellular antioxidant [41]. There are two types of glutathione, GSH and GSSG, and GSH is oxidized to GSSG by GPx with the detoxification of H_2_O_2_. GSSG is regenerated into GSH by glutathione reductase. Since total glutathione concentrations and the GSH/GSSG ratio are altered by oxidative stress, they are used as an indicator of oxidative stress.

In the present study, SOD and catalase activities were slightly lower in the DMC group than in the NOC group (Figure 4A,B). Increased H_2_O_2_ concentrations have been shown to inactivate SOD [42]. The decrease observed in SOD activity in the DMC group may be attributed to its inactivation by oxidative stress. The slightly reduction in catalase activity in the DMC group may suggest that liver tissues did not eliminate H_2_O_2_. However, SOD and catalase activities were higher in the DMP group than in the DMC group (Figure 4A,B), indicating the antioxidant potential in NEP for these antioxidant enzymes. On the other hand, GPx activity was significantly higher in the DMC group than in the NOC group (Figure 4C) and appeared to depend on increases in H_2_O_2_ concentrations by hyperglycemia and decreases in catalase activity. GPx activity was slightly lower in the DMP group than in the DMC group (Figure 4C). GPx detoxifies not only H_2_O_2_ but also organic peroxides, using GSH as a cofactor. Since the total glutathione concentration was lower in the DMP group (Figure 5B) and GSSG was higher in the DMP group (Figure 5D), GPx may not have functioned due to the lack of its cofactor.

The GSH/GSSG ratio and total glutathione, GSH, and GSSG concentrations did not significantly differ between the DMC and NOC groups (Figure 5), which is consistent with the ORAC of the liver tissues (Figure 3B). On the other hand, the GSH/GSSG ratio, total glutathione and GSH concentrations were significantly lower in the DMP group than in the NOC and DMC groups (Figure 5A–C), while GSSG concentrations were significantly higher in the DMP group than in the NOC and DMC groups (Figure 5D). Since the ORAC of liver tissues significantly increased in the DMP group (Figure 3B), NEP may exhibit radical scavenging activity instead of glutathione. A previous study demonstrated a significant increase in the GSH/GSSG ratio of kidney tissues in diabetic rats administered fractions of both polymers and oligomers, respectively, from proanthocyanidin extracted from persimmon peel [43]. Oxidative stress increases and antioxidant enzyme activities are altered under hyperglycemic conditions. However, conflicting findings have been reported in diabetic rats; namely, increases, decreases, and no changes in antioxidant enzyme activities [21,39,42,44,45].

Skeletal muscle fibers are classified into three basic types: SO, FG, and FOG types [46]. The nature of skeletal muscle is influenced by the proportions of the different types of muscle fibers. The SO type has many large mitochondria, high oxidase activity, and low glycolytic enzyme activity. It also has an excellent aerobic capacity. The FG type has a few small mitochondria, low oxidase activity, and high glycolytic enzyme activity. It also has an excellent anaerobic capacity. The FOG type has intermediate characteristics between SO and FG. There are no FG fibers in the soleus muscle of rats [47].

In the present study, the CSAs of all muscle fiber types in the EDL and soleus muscles were significantly lower in the DMC group than in the NOC group (Figure 7). Andersen et al. reported foot muscle atrophy in type 1 diabetes patients [48]. Bonnard et al. demonstrated that ROS production in streptozotocin-treated mice altered the structure of mitochondria and reduced their density [49]. Antioxidant enzyme activities and antioxidant concentrations were examined only in liver tissues and not in muscle fiber in this study; therefore, it is unclear if ROS is produced in muscle fiber. If ROS is produced in muscle fiber of the DMC group, it may induce structural alterations and decrease the density of mitochondria, which may have reduced metabolism and decreased the CSA of each muscle fiber. Reductions in the CSA of all muscle fiber types in the EDL muscle were significantly smaller in the DMP group than in the DMC group (Figure 7A). Bonnard et al. also reported that the normalization of glycemia and treatments with antioxidants decreased ROS production and restored mitochondrial integrity [49]. If antioxidant enzyme activities, such as SOD, and antioxidant concentrations increased in the muscle of the DMP group, this may have restored the structure of mitochondria, which may have promoted metabolism and increased the CSA of each muscle fiber type in the EDL muscle.

## 5. Conclusions

The present study examined the antioxidant potential of NEP against oxidative stress in the liver and muscle fibers in streptozotocin-induced diabetic rats. The administration of NEP increased the ORAC of liver tissues and SOD activity in diabetic rats. In addition, reductions in the CSA of EDL muscle fiber types were suppressed.

## Figures and Tables

**Figure 1 antioxidants-11-01555-f001:**
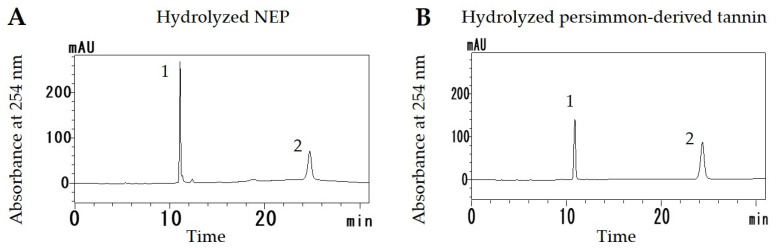
HPLC chromatograms of hydrolyzed non-extractable fraction of dried persimmon (NEP) (**A**) and persimmon-derived tannin (**B**).

**Figure 2 antioxidants-11-01555-f002:**
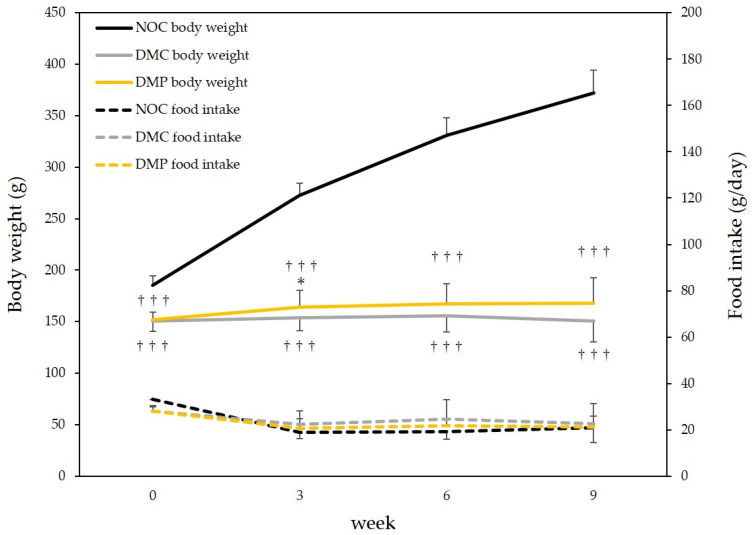
Changes in food intake and body weight in normal and diabetic rats. NOC: normal rats fed a control diet, DMC: diabetic rats fed a control diet, DMP: diabetic rats fed non-extractable fraction of dried persimmon (NEP) diet. Data are presented as the mean ± SD of each animal in each group. * *p* < 0.05 significantly different from the DMC group. ^†††^
*p* < 0.001 significantly different from the NOC group.

**Figure 3 antioxidants-11-01555-f003:**
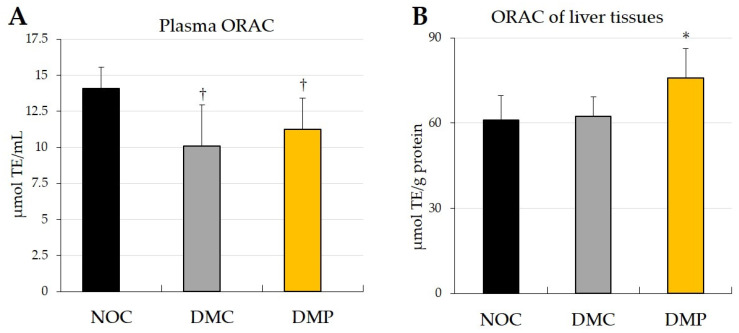
Effects of non-extractable fraction of dried persimmon (NEP) on the ORAC of plasma (**A**) and liver tissues (**B**) after 9 weeks in normal and diabetic rats. NOC: normal rats fed a control diet, DMC: diabetic rats fed a control diet, DMP: diabetic rats fed NEP diet. Data are presented as the means ± SD of each animal in each group. ^†^
*p* < 0.05 significantly different from the NOC group. * *p* < 0.05 significantly different from the NOC and DMC groups.

**Figure 4 antioxidants-11-01555-f004:**
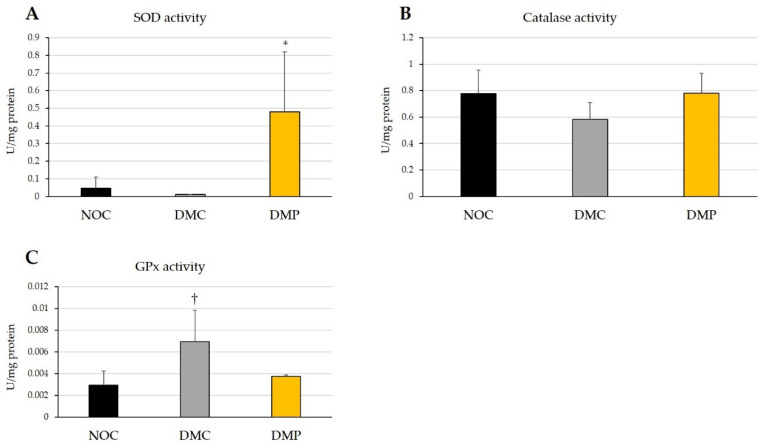
Effects of non-extractable fraction of dried persimmon (NEP) on the SOD activity (**A**), catalase activity (**B**), and GPx activity (**C**) of liver tissues in normal and diabetic rats. NOC: normal rats fed a control diet, DMC: diabetic rats fed a control diet, DMP: diabetic rats fed NEP diet. Data are presented as the means ± SD of each animal in each group. * *p* < 0.05 significantly different from the NOC and DMC group. ^†^
*p* < 0.05 significantly different from the NOC group.

**Figure 5 antioxidants-11-01555-f005:**
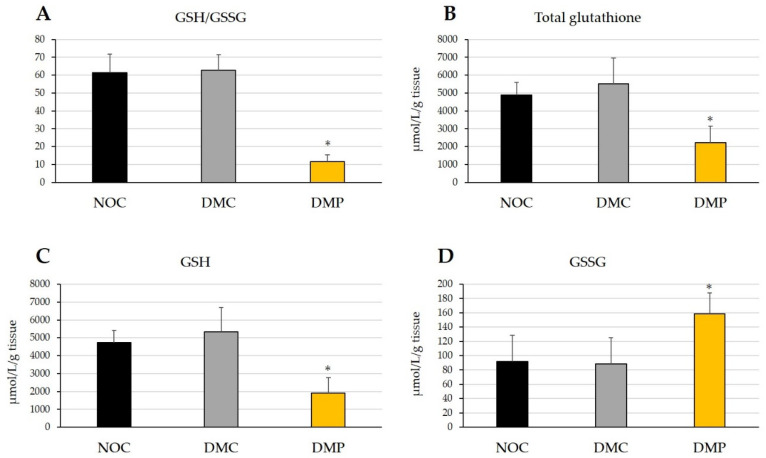
Effects of non-extractable fraction of dried persimmon (NEP) on the GSH/GSSG ratio (**A**), total glutathione concentrations (**B**), GSH concentrations (**C**), and GSSG concentrations (**D**) in liver tissues of normal and diabetic rats. NOC: normal rats fed a control diet, DMC: diabetic rats fed a control diet, DMP: diabetic rats fed NEP diet. Data are presented as the means ± SD of each animal in each group. * *p* < 0.05 significantly different from the NOC and DMC groups.

**Figure 6 antioxidants-11-01555-f006:**
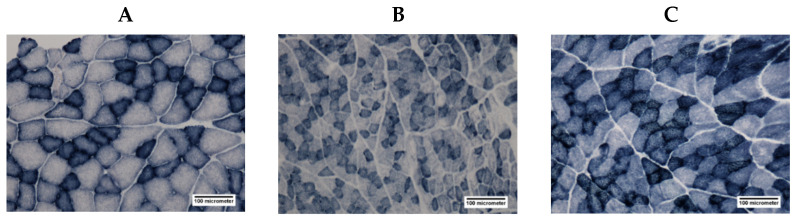
Photomicrographs of cross-sections through the extensor digitorum longus (EDL) muscle of the normal rats fed a control diet (NOC group) (**A**), diabetic rats fed a control diet (DMC group) (**B**), and diabetic rats fed the non-extractable fraction of dried persimmon (NEP) diet (DMP group) (**C**) stained with succinate dehydrogenase. (**A**–**C**) are shown at the same magnification, and the scale bar at the bottom right of each figure is 100 µm.

**Figure 7 antioxidants-11-01555-f007:**
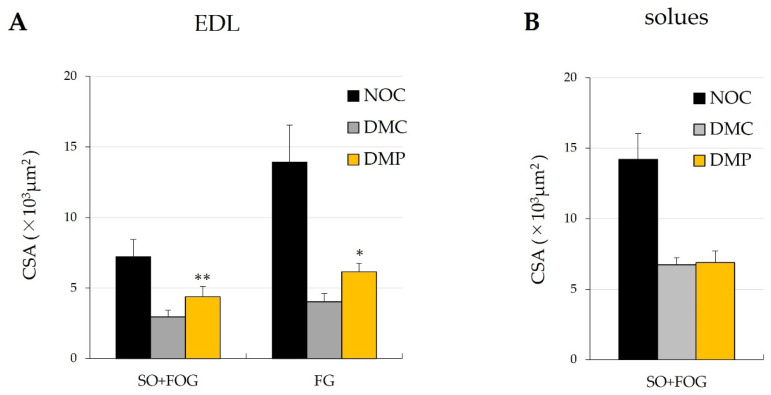
Cross-sectional area (CSA) of each fiber type in extensor digitorum longus (EDL) (**A**) and soleus (**B**) muscles in normal and diabetes rats. NOC: normal rats fed a control diet, DMC: diabetic rats fed a control diet, DMP: diabetic rats fed the non-extractable fraction of dried persimmon (NEP) diet. SO: slow-twitch oxidative type of muscle fiber, FOG: fast-twitch oxidative glycolytic type of muscle fiber, FG: fast-twitch glycolytic type of muscle fiber. Data are presented as the means ± SD of each animal in each group. ** *p* < 0.01 significantly different from the DMC group. * *p* < 0.05 significantly different from the DMC group.

**Table 1 antioxidants-11-01555-t001:** Compositions of diets (%).

Ingredients	NOC and DMC Groups Control Diet ^a^	DMP Group NEP Diet ^a^
Corn starch	39.75	39.75
Casein	20.0	20.0
α-Corn starch	13.2	13.2
Sucrose	10.0	10.0
Soybean oil	7.0	7.0
Mineral mix ^b^	3.5	3.5
Vitamin mix ^c^	1.0	1.0
L-cysteine	0.3	0.3
Choline bitartrate	0.25	0.25
Crystalline cellulose	5.0	-
NEP	-	5.0

^a^: The AIN-93G-modified basal diet without di-*tert*-butylhydroxytoluene was used. ^b^: Mineral mix AIN-93G. ^c^: Vitamin mix AIN-93G. NOC: normal rats fed an AIN-93G-modified basal diet, DMC: diabetic rats fed an AIN-93G-modified basal diet, DMP: diabetic rats fed a basal diet supplemented with 5% non-extractable fraction of dried persimmon (NEP) instead of cellulose.

## Data Availability

The data from this work are included in this article and may be obtained from the corresponding author.

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
