# Peer review of "Antioxidant Potential of Non-Extractable Fractions of Dried Persimmon (Diospyros kaki Thunb.) in Streptozotocin-Induced Diabetic Rats"

_antioxidants, 2022, doi:10.3390/antiox11081555_

Round 1

Reviewer 1 Report

It is a very organized and novel work.

There is some typist errors.

Author Response

Plese refer attacned file.

Author Response

Please refer attached file.

Reviewer 3 Report

The article is a well-written investigation on the potential antioxidant activity of the non-extractable fractions of dried persimmon on streptozotocin-induced diabetes rats.

The introduction provides a generalized background of the topic and gives some information for the readers to understand the general goal of the study. The methods used are appropriate to the aims of the study and there is sufficient information provided for a researcher to reproduce the experiments described. 

However, some adjustments must be taken. Here below I report my comments and my suggestions:

-       Why did you not include in the study normal rats fed with a basal diet supplemented with 5% NEP? 

-       Figure 1 can be improved: I suggest using the same y-axis to give also some perspective on the quantitative differences between hydrolyzed NEP and hydrolyzed persimmon-derived tannin

-       Also Figure 2 can be improved: if you want to keep body weight and food intake in the same figure, you can use different colors for the two parameters. 

-       From Figure 2, why NOC group has this high increase in Body weight?

-       Figure 3A: there is an error in the p-value as DMC group has only one  while both in the text and in the figure caption the p-value is p<0,01. Please correct it. 

-       Figure 4:  can you provide the dot plot? The differences between the groups are strange and the statistical analysis seems to be in contrast with graphical representation of the results. How cannot be a statistical reduction between DMC group and NOC group in Figure 4A? In the same way how can you observe a statistical difference between DMP and DMC but not between NOC and DMC?

-       Figure 4: The y-axes are all different. Can you express everything with U/mg protein?

-       Figure 5 and Figure 7: to better emphasizes your results I suggest indicating p<0,001 as ### instead of #. 

-       Figure 7: put the same y-axes for both graphs and enlarge legends as the legend squares are too little to identify which bar corresponds to each group. For Figure 7B: reduce the dimension of the bars and make it similar to the ones of 7A.

-       On the basis of the results, I suggest dampening a bit the conclusions of the work (line 56).

Author Response

Plese refer attached file.

Reviewer 4 Report

Presented study provides information about the antioxidant potential of non-extractable fractions of dried persimmon (Diospyros kaki Thunb.) on streptozotocin-induced diabetes in rodent model.

The topic is important, although papers related to beneficial effects of persimmon or its functional components (such as proanthocyanidins, carotenoids, tannins) on human health (against oxidative stress, hypertension, diabetes mellitus, and atherosclerosis) are already recorded. Thus, novelty is (to some extent) questionable. Still, the work is interesting and it may contribute to the overall insight into antidiabetic potential of dried persimmon.

Manuscript should be better prepared.

Please improve title: e.g. “Antioxidant potential of non-extractable fractions of dried persimmon (Diospyros kaki Thunb.) in streptozotocin-induced diabetic rats” or “Antioxidant potential of non-extractable fractions of dried persimmon (Diospyros kaki Thunb.) on streptozotocin-induced diabetes in Wistar rats”

Abstract - it is not necessary to write the abstract in numbered sections

Authors are also encouraged to provide a graphical abstract.

The introduction section should better present the current state of the research field. A more comprehensive review of the literature is needed given the large number of scientific studies conducted in this area.

 Materials and Methods

In Table 1instead of: “DMC group NEP diet” should be DMP group

Animal studies - three groups of rats were included in the study: normal rats fed an AIN-93G-modified basal diet (CLEA Japan Inc., To-137 kyo, Japan) (NOC group) and diabetic rats divided into two groups of six animals each, as follows: (i) diabetes rats fed an AIN-93G-modified basal diet (DMC group); (ii) diabetes rats fed a basal diet supplemented with 5% NEP instead of cellulose (DMP group). Control group of normal rats fed with basal diet supplemented with 5% NEP is missing - important for future experiments.

Important! Discussion – authors should refer to specific figure presented in Results section.

Figure 2 – please point out significant differences in body weight among NOC group vs. DMC group, and NOC group vs. DMP group. Also, a more detailed description of the image is required - it should be emphasized that an non-extractable fractions of dried persimmon has been applied in one group of rats.

Figure 3 (and other figures) – As in Fig. 2, a more detailed description of the image is required - it should be emphasized that an non-extractable fractions of dried persimmon has been applied in one group of rats.

Figure 6B - is the magnification the same as in Fig 6A, Fig 6C?

Many journal readers look only at displayed items (e.g. figures) without reading the main text of manuscript. Therefore, ensure that figures can stand alone from the text and communicate clearly your results (explain all abbreviations used in figures – NOC; DMC; DMP; etc).

Author Response

Please refer attached file.

Round 2

Reviewer 2 Report

Although the experimental work is not entirely satisfactory and it is hoped that it can be completed in future investigations, some sentences in the manuscript have been improved and therefore their presentation has been improved.

the manuscript can be published because it reports some interesting results on the subject under investigation, although not conclusive, and can provide ideas for further and more in-depth research.

Reviewer 3 Report

As the authors have responded to all my questions and made the necessary changes to the manuscript, I recommend accepting the paper.